# Overview of Gene Expression Dynamics during Human Oogenesis/Folliculogenesis

**DOI:** 10.3390/ijms25010033

**Published:** 2023-12-19

**Authors:** Bastien Ducreux, Lucile Ferreux, Catherine Patrat, Patricia Fauque

**Affiliations:** 1Université Bourgogne Franche-Comté-Equipe Génétique des Anomalies du Développement (GAD) INSERM UMR1231, 2 Rue Angélique Ducoudray, F-21000 Dijon, France; bastien.ducreux@outlook.fr; 2Faculty of Medicine, Inserm 1016, Université de Paris Cité, F-75014 Paris, France; lucile.ferreux@aphp.fr (L.F.); catherine.patrat@aphp.fr (C.P.); 3Department of Reproductive Biology-CECOS, Aphp.Centre-Université Paris Cité, Cochin, F-75014 Paris, France; 4Laboratoire de Biologie de la Reproduction-CECOS, CHU Dijon Bourgogne, 14 Rue Gaffarel, F-21000 Dijon, France

**Keywords:** folliculogenesis, maturation, oocyte, RNA-seq, transcriptome

## Abstract

The oocyte transcriptome follows a tightly controlled dynamic that leads the oocyte to grow and mature. This succession of distinct transcriptional states determines embryonic development prior to embryonic genome activation. However, these oocyte maternal mRNA regulatory events have yet to be decoded in humans. We reanalyzed human single-oocyte RNA-seq datasets previously published in the literature to decrypt the transcriptomic reshuffles ensuring that the oocyte is fully competent. We applied trajectory analysis (pseudotime) and a meta-analysis and uncovered the fundamental transcriptomic requirements of the oocyte at any moment of oogenesis until reaching the metaphase II stage (MII). We identified a bunch of genes showing significant variation in expression from primordial-to-antral follicle oocyte development and characterized their temporal regulation and their biological relevance. We also revealed the selective regulation of specific transcripts during the germinal vesicle-to-MII transition. Transcripts associated with energy production and mitochondrial functions were extensively downregulated, while those associated with cytoplasmic translation, histone modification, meiotic processes, and RNA processes were conserved. From the genes identified in this study, some appeared as sensitive to environmental factors such as maternal age, polycystic ovary syndrome, cryoconservation, and in vitro maturation. In the future, the atlas of transcriptomic changes described in this study will enable more precise identification of the transcripts responsible for follicular growth and oocyte maturation failures.

## 1. Introduction

Obtaining a mature oocyte is a lengthy and complex process. It results from two processes, folliculogenesis and oogenesis, which set the stage for early embryonic development. Interestingly, these multistep procedures are characterized by specific transcriptional states and are associated with notable transcriptomic transitions. To date, there is still much we do not know about the tight regulation of oogenesis.

The long journey of an oocyte is initiated in the primordial germ cells during fetal life when oogonia enter meiosis (approximately at week 9 in humans) and is subsequently arrested at the diplotene stage of prophase I (germinal vesicle stage, GV) [1,2]. The nuclear maturation is stopped for several years and resumes after puberty at each menstrual cycle, when a surge of luteinizing hormone triggers the final maturation of a dominant follicle recruited to develop until the oocyte reaches the metaphase of meiosis II (MII) stage. The oocyte has then completed both cytoplasmic and nuclear maturation. This maturation is manifested by transcriptional and physiological/morphological changes that should provide optimal conditions for fertilization and the transport of essential maternal transcripts that will control the first embryonic divisions [3,4,5]. Concomitantly, oocyte nuclear maturation is also conditioned by two other biological modifications: organelle maturation, which situates mitochondria/ribosomes/endoplasmic reticulum/cortical granules and the Golgi apparatus, and epigenetic maturation, because important epigenetic reprogramming takes place at the same time (de novo methylation, histone modifications and exchanges) [6]. Thus, oocyte growth and maturation are the main factors determining early embryonic competence.

In the pre-antral follicles, the chromatin is decondensed, making the transcriptional activity high and fitting the needs required by the follicle to grow [7,8]. As the tertiary follicle develops, gradual condensation of chromatin occurs, and the transcriptional activity is finally stopped when oocytes reach the GV breakdown (GVBD) and could initiate drastic mRNA degradation, as found in the mouse model [9,10]. The degradation of maternal transcripts is needed to pave the way for the embryonic transcriptome and occurs in two big waves. Maternal decay (M-decay) takes place during final oocyte maturation and is thought to eliminate 30–50% of maternal factors, while zygotic decay (Z-decay) occurs after fertilization and clears the remaining material in such a way that approximately 80% of polyadenylated (poly(A)) mRNAs stored at the GV stage are not present in the eight-cell mouse embryo [10,11,12,13,14]. This degradation is notably governed by RNA-binding proteins, microRNAs, nonsense-mediated decay factors, or siRNA [15,16,17,18]. Reaching the MII stage does not guarantee that the embryo will fully develop, and oocyte competence should be decoded from their transcriptome [5,19,20]. The key process in oocyte maturation that will determine embryo quality may lie (i) in the selective degradation and conservation of maternal transcripts from the fully grown GV to MII stage and (ii) in the appropriate and timely regulation of a set of specific transcripts at each stage of oocyte development.

Advances in the knowledge of oocyte maturation and folliculogenesis may also be beneficial for the success of fertility preservation. It may lead to the optimization of in vitro maturation and culture processes in fertility clinics by identifying key factors conditioning the favorable outcome of oocyte maturation. In addition, it may lead to a better understanding of what causes deficient maturation, because excessive transcript accumulation and defects in the degradation machinery were shown to be associated with embryonic genome activation failure [11,15,21,22]. Moreover, the development of RNA-seq techniques has expanded our knowledge of the transcriptome-wide landscape of tissues and even single cells (scRNA-seq). Capturing polyadenylated RNA in a sample provides a snapshot of the existing mRNA population and reflects its gene expression identity. Over the last few years, RNA-seq has been used to profile the transcriptome of growing and maturing oocytes, but no studies have comprehensively aggregated the resulting data.

In the current review, we comprehensively evaluated these data to better identify key processes involved in human oocyte growth and maturation. We first applied a Gene Ontology and a pseudotime analysis to identify the fundamental transcriptomic requirements of the oocyte at any stage of folliculogenesis and the resultant mRNA regulation. Secondly, we performed a meta-analysis of studies comparing fully grown GV and MII oocytes to decode the accumulation/degradation of maternal transcripts occurring during this final maturation. In closing, we evaluated whether the transcripts supposedly having a role in the oocyte growth and maturation processes are sensitive to environmental factors, which could help us better understand the origin of oocyte maturation defects.

## 2. Results

### 2.1. Folliculogenesis

Global observation via principal component analysis (PCA) of transcriptomes from oocytes originating from primordial to antral follicles indicated overall differences between the different groups, but there was little difference between primordial and primary follicles, suggesting high transcriptional profile proximity for those cells (Figure 1A). The accumulation of maternal transcripts in the oocyte throughout folliculogenesis was confirmed, but there was no significant variation from the primary to the secondary follicle stage (Figure 1B).

We carried out a gene ontology over-representation analysis on genes differentially expressed between one stage and all others to reveal the biological signatures of oocytes from each follicle type (Figure 1C, Appendix A).

Oocytes of primordial follicles are characterized by the upregulation of genes involved in cytoplasmic translation, chemotaxis, regulation of cell adhesion, and regulation of epithelial cells, while the cell cycle phase transition is downregulated. Primary follicle oocytes do not display strong transcript identity compared to oocytes from other stages of follicle development, since few pathways are significantly downregulated and there is low biological function linked to oocyte development. Oocytes of secondary follicles manifest a rise in aerobic respiration, mitochondrial ATP synthesis, and oxidative phosphorylation, while RNA processes (mRNA processing, RNA splicing) tend to be slowed down. Finally, oocytes of antral follicles reveal a pattern that is opposite to primordial follicles (downregulation of cytoplasmic translation, chemotaxis, epithelial cell regulation, upregulation of cell cycle transition regulation) and secondary follicles (downregulation in aerobic respiration, mitochondrial ATP synthesis, and oxidative phosphorylation). Oocytes of antral follicles are also characterized by higher expression of genes involved in recombinational repair.

Folliculogenesis is likely a dynamic process since oocytes in follicles undergo continuous transcriptomic modifications and are not arrested at definite stages. As such, we performed a pseudotime analysis to temporally assess which genes show variation of expression in the window of folliculogenesis and to characterize the activation or repression behavior of these transcripts throughout follicular development. We first applied PHATE dimension reduction to normalized counts and discerned a transcriptomic trajectory (Figure 2A). We observed that primordial and primary follicles were almost indistinguishable in the first two dimensions (Figure 2A). A heatmap comparing differentially expressed genes all in one stage versus all other stages confirmed this observation considering that hierarchical clustering hardly separates primordial from primary follicles, revealing their transcriptomic proximity (Appendix A). We also observed that in the same follicle stage group, the transcriptome could be temporally different. Specifically, some antral follicles were close to secondary follicles, while the remaining antral follicles formed a clearly separated cluster later in the follicle developmental trajectory (Figure 2A), which turned out to be related to a patient effect (Patients E and G in Appendix A). The patients included in this study were notably healthy or suffering from reproductive pathologies of varying degrees of severity.

Then, we generated a list of 6552 transcripts whose expression varied temporally during folliculogenesis according to our pseudotime analysis (FDR < 0.05, logFC > 1) (Appendix A). The heatmap in Figure 2B highlights the sequential waves of transcriptional activity modifications from the primordial to the antral stage (Figure 2B). Successive activation and repression of these genes may drive folliculogenesis and regulate the transition from early to late stages, which is particularly the case for transcription factors (Appendix A). Most of the 6552 transcripts are highly expressed inside the primordial and primary follicles and show lower expression in the following stages (3903 transcripts) (Figure 2B) but they are not enriched for a specific gene ontology. The transition to a secondary follicle oocyte is marked by the activation of a small set of genes (152) related to peptidyl-tyrosine modification and kinase activity, while final follicle oocyte growth is associated with the upregulation of 2497 transcripts enriched in cell cycle pathway genes (Figure 2B). The top five genes for each follicle stage group (primordial/primary, secondary, and antral) are displayed in Figure 2C, and their function is described in Table 1.

### 2.2. Oocyte Maturation

Re-analysis of individual studies

Standard re-analysis of six studies comparing the transcriptome of fully grown GV and MII oocytes obtained after controlled ovarian stimulation confirmed the existence of intense modulation of maternal transcript abundance [24,25,26,27,28,29]. First, a clear separation between fully grown GV and MII oocytes was observed in all datasets using dimension reduction (PCA method) (Appendix A). Secondly, the number of differentially expressed genes between GV and MII oocytes was high in all datasets, with an average of 33% (9–51%) of transcripts undergoing modification between the two stages (Figure 3A). According to several independent studies, the main pathways modified were related to mitochondrial processes (downregulated in MII compared to fully grown GV), cytoplasmic translation, and chromatin organization (upregulated in MII compared to fully grown GV) (Appendix A). Correlations in all transcript expressions were high between datasets, but there were some differences, likely due to technical variation, justifying the need for a meta-analysis (Figure 3B). The six RNA-seq datasets showed high correlation in all expressed gene counts (Pearson’s correlation coefficient > 0.75, Figure 3B), but as seen on PCA on merged datasets, the various techniques have an influence on transcript abundance even after batch correction (Appendix A).

Meta-analysis

To eliminate parameters likely to influence biological effects (such as age or health condition) and increase statistical power, we conducted a meta-analysis of differentially expressed genes (DEGs) between fully grown GV and MII oocytes. In total, the six datasets included 87 fully grown GV and 70 MII oocytes, collected from patients ranging from young to advanced maternal ages (18–44 yo). The random effects model meta-analysis identified 2991 DEGs with consistent modifications across all six datasets (Figure 3C, Appendix A). While most were downregulated in MII oocytes (61%) (Figure 3C), some displayed a higher level of transcripts in MII oocytes. Upregulated pathways were related to histone modification, nuclear maturation, and RNA processes, while downregulated pathways were related to mitochondrial processes (Figure 4). Transcript annotation revealed that DEGs were mainly protein-coding genes and, in small proportions, transposable elements or long non-coding RNAs (lncRNA) (Appendix A). Long terminal repeats (LTR) were mostly upregulated in MII, while long interspersed nuclear elements (LINE) elements were downregulated (Appendix A).

We evaluated the extent of mRNA degradation during the transition from fully grown GV to MII oocyte stage according to the meta-analysis results. A substantial proportion of the transcripts was highly degraded (>70% loss) (Figure 5A). The mean degradation per transcript was 53% for downregulated transcripts (Figure 5B). Between fully grown GV and MII stages, there was a significant drop in total maternal mRNA quantity, which could be evaluated at 31% (Figure 5C). The degradation concerned transcripts from all chromosomes, whereas mtDNA gene expression was increased in MII oocytes (Figure 5D).

### 2.3. Influence of the Environment on Human Oogenesis/Folliculogenesis

We identified transcripts differentially expressed during folliculogenesis or the GV-to-MII transition; these transcripts are also putative transcripts susceptible to environmental stressors occurring in the window of oocyte development (maternal age, polycystic ovary syndrome (PCOS), in vitro maturation (IVM), cryopreservation). A list of these genes is available in Table 2 and a summary of studies on the factors evaluated can be found in Appendix A.

## 3. Discussion

The oocyte is the largest human cell and one of the richest in mRNA quantity [30]. It determines early embryonic competence and its transcriptome contains many secrets waiting to be revealed [31]. In this study, we comprehensively gathered scRNA-seq datasets from human oocytes at multiple stages during the long process of oocyte growth and maturation and decrypted the transcriptomic reorganization required to ensure that the oocyte becomes fully competent. We believe it is important to focus on human oocyte regulation exclusively, as previous studies comparing human oocyte maturation with other mammalian species revealed very few overlaps in the individual transcripts modified, even if major biological processes activation/repression were similar [18].

Folliculogenesis and oocyte maturation imply specific regulation during oocyte development. First, variation in chromatin accessibility determines the activity of transcription, which is high within early follicles but absent after GV breakdown [7,32]. Secondly, transcription factor activity shapes the oocyte transcriptome during folliculogenesis by upregulating or downregulating specific transcripts to allow the transition from a primary to a pre-ovulatory follicle oocyte [33]. Thirdly, selective degradation or translation of maternal transcripts is a prerequisite for cytoplasmic maturation, leading to a fertilizable MII oocyte that can ensure the first embryonic divisions [14].

Our chronological gene ontology comparison analysis revealed signature pathways activated and repressed in oocytes from the different follicle stages. The primordial stage may reflect strong communication between the oocyte and its surrounding cells, manifested by an upregulation of genes related to cell adhesion, epithelial cell proliferation, and collagen fibril organization pathways, which tend to be reduced afterwards. Communication with the surrounding cells, such as granulosa, plays a pivotal role in oocyte growth via the action of paracrine factors [34]. Oocytes mobilize specific transcripts such as *GDF9*, *TGFB1*, and *ACTN1*, to act on granulosa cell proliferation and maintain gap junctions [35,36]. Moreover, downregulation of cytoplasmic translation was seen in antral follicles while the highest activity was observed in primordial follicles, which corroborates the findings that progressively maternal mRNAs are not translated but stored until they are degraded or translated after meiosis resumption following GVBD. Oocytes from primordial follicles also show downregulation of cell cycle pathways which agrees with their arrest at the diplotene stage of meiosis I until their activation [37]. In the antral follicle, cell cycle phase transition pathways are upregulated, possibly correlating with the acquisition of meiotic competence to sustain proximate oocyte maturation [38]. One of our conclusions would be that the oocytes at the primary follicle stage do not display a highly specific transcriptomic profile compared to the primordial follicle. The transition between both stages is likely to be mRNA quantity-specific rather than pathway-specific, meaning that primary follicle oocytes accumulate mRNA drastically but not selectively. This primary-to-primordial transition may be driven by a specific regulatory network involving transcription factors and long non-coding RNAs according to previous evidence [23,39]. Then, the transition from an early follicle oocyte to a secondary follicle oocyte is marked by a switch in energy consumption. Activation of mitochondrial ATP synthesis, aerobic respiration, and oxidative phosphorylation sustain major structural and biochemical evolution, which are energy-intensive processes [40]. Furthermore, mRNA processing and notably splicing are slowed down in secondary follicle oocytes compared to other stages. Finally, oocytes in antral follicles turn off energy activity possibly passively (downregulation of aerobic respiration and oxidative phosphorylation pathways) because of the difficulty of accessing O_2_ in a restricted micro-environment, according to mathematical modelling in humans [41,42,43]. We also observed that oocytes in antral follicles show an increased ability to repair DNA alterations compared to other stages (upregulation of recombinational repair pathway), which is confirmed by previous observations [44].

Taking advantage of the single-cell nature of the Zhang et al. dataset [23], we then applied trajectory analysis along folliculogenesis to characterize continuous transcriptomic changes. We identified a bunch of genes showing significant variation in expression from primordial-to-antral follicle oocyte development and characterized their temporal regulation. This revealed that a large majority of them were highly upregulated during the early follicle stages, seemingly biologically unspecific as no Gene Ontology showed enrichment for them. This confirms our hypothesis of an unspecific accumulation of transcripts during the primary follicle stage. Surprisingly, only about a hundred folliculogenesis variable genes show a peak of expression in the window of the secondary follicle stage, which appeared to be related to peptidyl-tyrosine modification and regulation of kinase (notably the MAPK cascade) and transferase activity. Following transcriptional waves of upregulation may involve the preparation for meiosis resumption in the antral follicle oocyte.

Furthermore, we observed that depending on the patients, the transcriptome dynamic displayed different kinetics of progression. Differences between patients in the previously published dataset used herein could stem from the inclusion of patients with particular pathological conditions (cervical cancer, endometrial cancer, lymphoma), which may affect reproductive functions. This could lead to unwanted bias in the interpretation of the results as a small number of patients for each follicle stage were included.

Next, we better characterized the downfall undergone by maternal mRNAs during late oogenesis. We found an average decline of 30% in total mRNA from fully grown GV to the MII stage across studies. Previous studies indicated that 60% of maternal transcripts are degraded from the fully grown GV to MII stage, using mass measures [45,46]. The difference observed may be a direct consequence of the fact that mRNAs with short polyA tails are not captured by the employed sequencing technologies. Therefore, our analysis mirrors the biological pathways that are recruited in a timely manner (mRNAs with long polyA tails on the verge of being degraded or translated) rather than how degraded all maternal transcripts are. However, this approach displays the selective downregulation of specific transcripts. Indeed, transcripts associated with energy production and mitochondrial functions were extensively downregulated, while those associated with cytoplasmic translation, histone modification, meiotic processes, and RNA processes were conserved. The downregulation of mitochondrial processes is highly conserved across mammals and may be a protective system with respect to oxidative stress via ROS production and the inability of the early embryo to protect against these stressors as reported in rhesus monkey, mouse, and cow models [18,47]. Concomitantly, MII oocytes should have already synthesized sufficient mtDNA to support key developmental events such as epigenetic programming [48]. The new meta-analysis we proposed found that genes involved in mobilization of cytoplasmic translation, histone modification, meiotic processes, and RNA processes are upregulated in MII oocytes, which are metabolic pathways crucial for the multiplicity of maturations (cytoplasmic-nuclear-epigenetic) required by the oocyte.

Interestingly, the transition from fully grown GV to MII oocytes displayed upregulation of a substantial number of transcripts, whereas oocytes are thought to be transcriptionally quiescent. This could be explained by several hypotheses. First, as explained, the scRNA-sequencing methods used in the meta-analyzed studies captured mRNA with long polyA tails, thus reflecting the amount of mRNA that can be recruited for translation and not silenced or set to degrade mRNAs with short polyA tails. Some transcripts are still enrolled for translation during the GV-MII transition via polyA lengthening, showing them to be upregulated in MII oocytes. A second hypothesis that still needs investigation is that transcriptional quiescence after the GVBD stage is not total [49,50]. This theory is contrasted by a third hypothesis, only proven in *C. elegans*, assuming that oocytes after GVBD cannot synthetize mRNA but can accumulate transcripts from surrounding somatic cells, supposedly via extracellular vesicles [51].

Furthermore, we intended to re-analyze studies comparing failed-to-mature oocytes versus mature oocytes. In Pietroforte et al., the authors did not identify any DEG between both groups, while the low number of samples in Li et al. (three oocytes per group) would be too low to detect expression differences with satisfactory power [52,53]. Then, it remains crucial to assess the biological reasons behind oocyte maturation failure and whether transcriptomic abnormalities would be responsible for it. In some cases, defects in the maternal mRNA degradation machinery would be involved, notably an alteration in the expression of *Btg4* or *Msy2* in mice [22,54,55]. This would lead to the unwanted maintenance of certain categories of transcripts detrimental to oocyte fertility or embryonic development.

In addition, throughout growth and maturation, oocytes are exposed to multiple environmental stressors, which can alter the sequential gene expression changes governing oocyte development [56]. Our identification of transcripts expressed along oocyte development that are highly susceptible to those factors is of major importance, as they may be the key to understanding the apparition of oocyte defects and deficiencies and overcoming these outcomes [27,28,29,57,58,59,60,61,62,63,64,65,66,67]. For oocyte in vitro maturation and cryoconservation, regaining expected levels of expression of those transcripts by adapting assisted reproductive protocols may become a path of research to increase mature oocyte rates in the future. For age and PCOS, identification of those transcripts is also important to understand the precise underlying mechanisms. In future studies, it would be highly interesting to observe transcriptomic modifications in the same patient from the primordial follicle stage and compare the dynamic of transcriptional changes between several donors, particularly under pathological conditions (pathology versus control).

To conclude, the oocyte transcriptome follows a dynamic that is tightly controlled. Overall, we have decoded the regulatory events of maternal mRNAs that lead the oocyte to grow and undergo multiple types of maturation, which we synthetized in Figure 6. The atlas of transcriptomic modifications described here will facilitate the precise identification of the transcripts involved in the failure of oocyte growth and maturation in future research.

## 4. Materials and Methods

### 4.1. Population

To examine the transcriptomic modifications experienced by the oocyte along human folliculogenesis, we included datasets comprising human primary oocytes from primordial to antral follicles (at least two different stages). To examine the transcriptomic modifications experienced by the oocyte throughout human oogenesis, we included datasets comprising human immature fully grown GV (prophase I arrested), MI (metaphase of meiosis I), and MII oocytes collected from controlled ovarian hyperstimulation protocols. Inclusion was also conditional on the sequencing method, which had to be single-oocyte RNA-seq. Only single-denuded oocytes were considered, meaning that oocytes surrounding cells were excluded from this review. Oocytes recovered in all clinical contexts were included, whether they were collected from donors or patients seeking infertility treatment, from in vitro or in vivo maturation, and at all maternal ages. No restrictions were placed on the publication date of datasets. A summary of the included studies with information on population characteristics, study design, and sequencing protocol is presented in Table 3.

### 4.2. Dataset Collection

All the datasets used in this study are available in public repositories. Raw data (.fastq) were downloaded from European Nucleotide Archive accessions: PRJNA666614, PRJNA508772, PRJNA377237, PRJNA811110, PRJNA701233, PRJNA690226 and PRJNA421274.

### 4.3. Dataset Processing

For each dataset, adapters and low-quality sequences were trimmed using TrimGalore! V0.6.6. Next, paired-end alignment was performed onto human reference genome (hg38) with STAR v2.7.9a, reporting randomly one position and allowing 6% of mismatches. Transposable element (TE) annotation was downloaded from RepeatMasker and joined with Gencode v19 gene annotation [68] to measure their expression level as TEs are crucial in the regulation of gametes and embryos gene expression. Finally, gene expression was quantified with featureCounts v2.0.1.

All datasets were processed following a standard procedure. Genes with low counts were removed with the filterByExpr function from the edgeR R package. Counts were normalized using the limma-voom method. When clinical information was available, we used duplicateCorrelation to account for samples from the same patients (considered as technical replicates).

For comparison of folliculogenesis stages (oocytes from primordial, primary, secondary, and antral follicles), differential analysis was performed between oocytes from one stage versus all stages using limma (differentially expressed if FDR < 0.05 and FC > 2 or FC < 0.5). This analysis was designed to identify a specific signature for each follicular stage. Because folliculogenesis is likely a dynamic process, a pseudotime analysis was implemented using tradeSeq using PHATE reduction to find differentially expressed genes throughout folliculogenesis rather than separating oocytes into distinct clusters [69,70]. This will identify precisely in time which genes are activated and repressed in oocytes during the development of follicles, from the primordial to the antral stage. The advantage of PHATE reduction is that it relies on diffusion dynamics and is thus particularly suitable to assess developmental trajectories such as oocyte growth.

For the comparison between GV, MI, and MII oocytes, differential analysis was performed between stage groups for each independent dataset using limma. Maternal age was incorporated as a covariate when available. Genes were considered differentially expressed when FDR < 0.05 and FC > 2 or FC < 0.5.

All PCAs in this article were performed using PCAtools on normalized expression (log2(cpm + 1)) removing the 10% genes with the lowest variance. Pathway analyses were run with clusterProfiler using the enrichGO function. For the purpose of visualization, independent datasets were corrected for batch effects using ComBat. All heatmaps were created with the pheatmap R package.

### 4.4. Meta-Analysis

A meta-analysis between all datasets comparing fully grown GV, MI, and MII oocytes was performed with MetaVolcanoR, which relies on the metafor package [71,72]. We summarized the expression fold change for each gene across the six independent datasets, implementing a random effects model using effect size (log2FC) and raw *p*-values from each dataset as input. The resulting estimation of a summary *p*-value for each gene indicates the probability of the summary expression fold-change being equal to zero. The advantage of the random effects model is the consideration of heterogeneity between studies (between-study effect variance) in addition to the within-study effect variance. For each gene, we considered significant expression change with an adjusted summary *p*-value < 0.05 and consistency (same sign of log2FC) in all studies.

Degradation or accumulation of mRNA quantity for each transcript through final oocyte maturation was calculated by converting the summary log(fold-change) to the percentage difference of normalized expression between baseline and the compared group. Transcripts with higher level of transcripts in MII than in fully grown GV were considered stable due to the assumed transcriptional quiescence of oocytes during the GV-to-MII transition (difference in polyadenylation and not quantity).

Because of the low number of studies that included MI oocytes (*n* = 2) and their reduced sample size (respectively, 6 and 7 MI), results including the comparison of MI with fully grown GV and MII oocytes are not discussed further (high risk of bias).

### 4.5. Influence of the Environment on Human Oogenesis/Folliculogenesis

In a previous review on the transcriptomic integrity of human oocytes used in assisted reproductive techniques (ARTs), we discussed the biological mechanisms likely to be altered in the context of ART interventions, maternal aging, a suboptimal lifestyle, or reproductive health issues [56]. We believe it would be highly interesting to identify oocyte growth and maturation-specific transcripts dysregulated by those factors. Thereby, available lists of DEGs from each study discussed in our previous review were retrieved and cross-checked for each environmental factor considered (endometriosis, PCOS, maternal age, smoking, body mass index (BMI), ovarian stimulation, IVM, and cryopreservation) to find common transcripts dysregulated between at least two studies. Those lists were in turn cross-checked with lists of DEGs identified in our folliculogenesis and oocyte maturation differential expression analyses. Previous studies on granulosa and cumulus cells were not considered here.

## Figures and Tables

**Figure 1 ijms-25-00033-f001:**
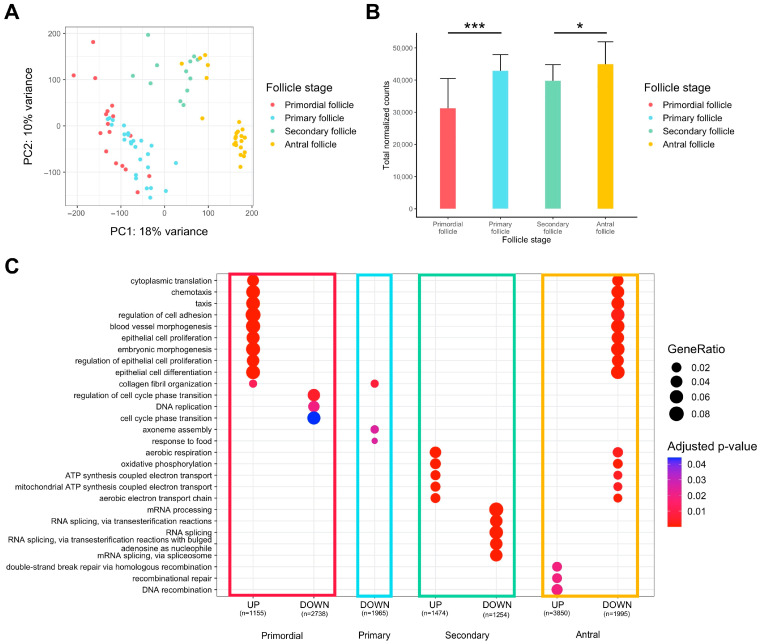
Differential expression analysis of transcriptomic signature changes along the different stages of folliculogenesis (dataset from Zhang et al. (2018) [23]). (**A**) Principal component analysis of the whole expression dataset removing the 10% genes with the lowest variance. (**B**) Comparison of the normalized read counts total between the different follicle stages (significance assessed with a *t*-test, * *p* < 0.05, *** *p* < 0.005). (**C**) Comparative Gene Ontology enrichment analysis according to the up- or downregulation of transcripts between all four follicle stages. GeneRatio corresponds to the fraction of differentially expressed genes (DEGs) found in each Gene Ontology set.

**Figure 2 ijms-25-00033-f002:**
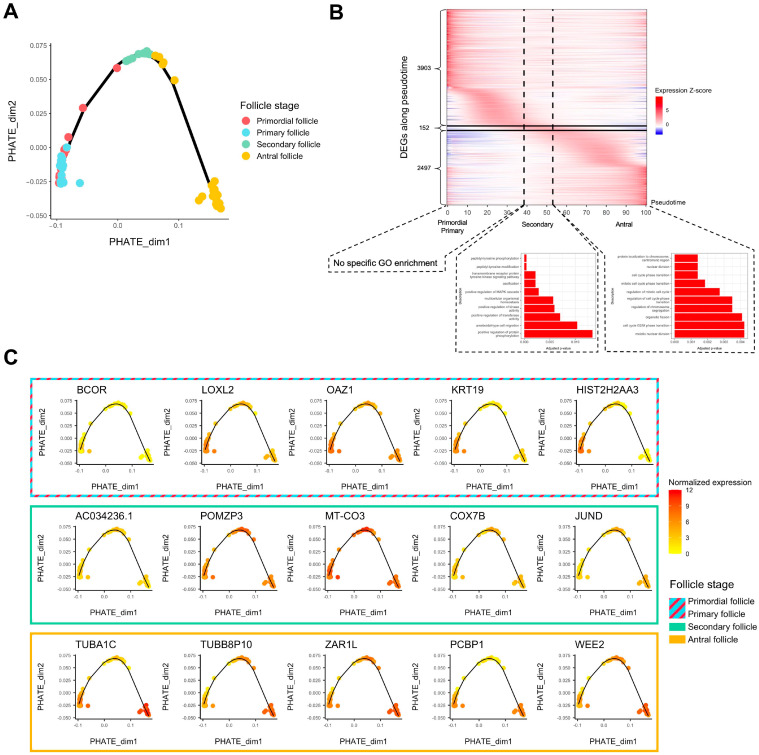
Trajectory analysis of transcriptomic changes along folliculogenesis. (**A**) PHATE dimensionality reduction analysis of the whole expression dataset. (**B**) Heatmap of inferred expression of the most variable significant genes along folliculogenesis (6552 genes showing average expression changed by at least logFC > 1 within the oocyte developmental trajectory). Pseudotime is a metric that could be interpreted as a timing distance between one cell and its precursor cell and helps identify the ordering of cells along a lineage based on their gene expression profile. In this analysis, pseudotime 0 represents oocytes in the beginning of the window of the oocyte developmental trajectory (early follicles), while pseudotime 100 represents oocytes in the end of the window (antral follicles). Vertical lines delimit oocytes from the primordial/primary, secondary, and antral follicle groups. Horizontal lines delimit significantly differentially expressed genes along folliculogenesis reaching their peak expression during the primordial/primary (3903), secondary (152), and antral follicle groups (2497). (**C**) Variation of expression, for each follicle stage group, of the top 5 genes differentially expressed along folliculogenesis reaching their peak expression at this stage.

**Figure 3 ijms-25-00033-f003:**
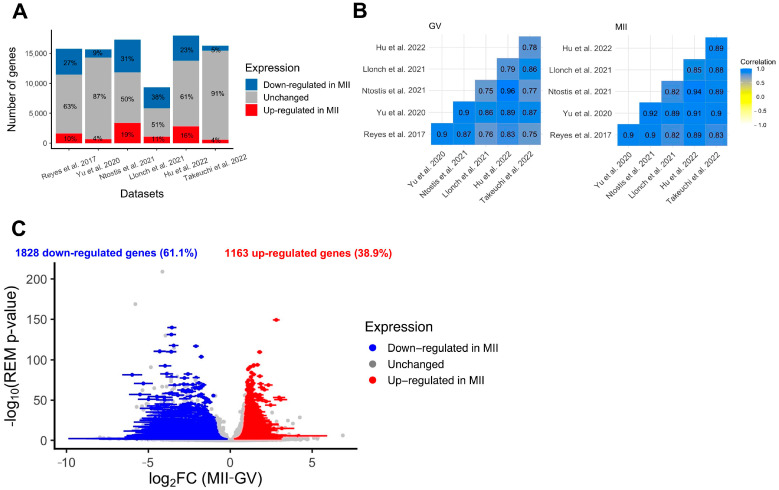
Differential expression analysis of genes during the GV-to-MII oocyte transition. (**A**) Results of individual differential expression analysis for the six datasets evaluated in this study. Numbers represent the percentage of genes upregulated, downregulated, or stable in MII versus GV oocytes. (**B**) Correlation of the normalized expression of all expressed genes for the six datasets evaluated in this study, respectively, for GV and MII oocytes. (**C**) Volcanoplot of the meta-analysis result for the six datasets evaluated in this study. A random effects model based on effect size (log2FC, difference of expression between MII and GV oocytes) and *p*-values from each dataset was applied. One gene is represented by one dot. A total of 2991 significant DEGs were identified (summary adjusted *p*-value from the random effect model < 0.05 and sign of log2FC consistent in all 6 studies). Blue dots represent significant DEGs downregulated in MII oocytes compared to GV. Similarly, red dots represent significant DEGs upregulated in MII oocytes compared to GV oocytes. The interval confidence of the summary log2FC is represented by the bars associated with the dots for each DEG [24,25,26,27,28,29].

**Figure 4 ijms-25-00033-f004:**
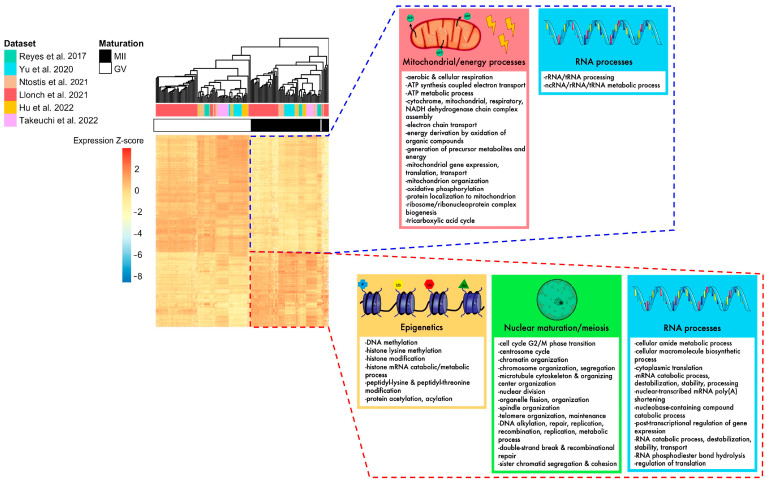
Summary of biological pathways enriched for DEGs identified in the meta-analysis, according to their up- or downregulation. The heatmap of DEGs identified with the meta-analysis shows that MII and GV oocytes are correctly segregated according to their expression for these genes. Mitochondrial processes, energy processes, and some RNA processes were downregulated in MII oocytes (framed in blue), while nuclear maturation/meiosis, epigenetics, and some other RNA processes were upregulated in MII oocytes (framed in red) [24,25,26,27,28,29].

**Figure 5 ijms-25-00033-f005:**
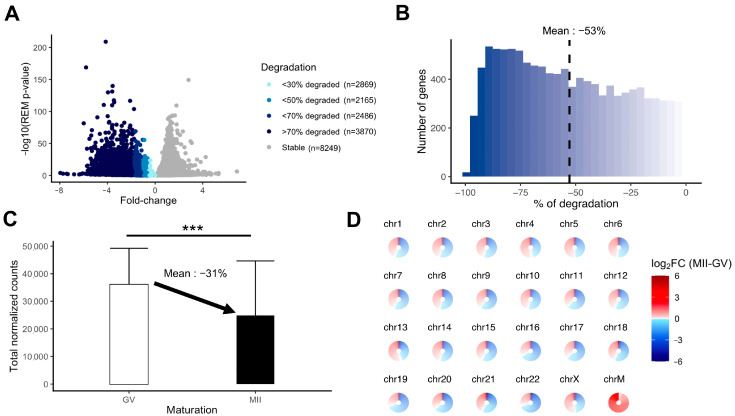
Measurement of the degradation of the maternal mRNA content in MII oocytes. (**A**) Volcanoplot of the meta-analysis result (MII versus GV) colored by the intensity of degradation between GV and MII stages (stable, <30% of degradation, <50% of degradation, <70% of degradation, or >70% of degradation). (**B**) Histogram focusing on the extent of degradation in downregulated transcripts in MII versus GV oocytes. (**C**) Comparison of total normalized counts between the GV and MII oocytes (significance assessed with a *t*-test, *** *p* < 0.005). (**D**) Assessment of degradation by chromosome. Each circle is divided into sections according to the number of genes on the chromosome and each section, representing one gene, is colored by the summary log2FC resulting from the meta-analysis result (MII versus GV) corresponding to the gene. ChrM: mitochondrial chromosome.

**Figure 6 ijms-25-00033-f006:**
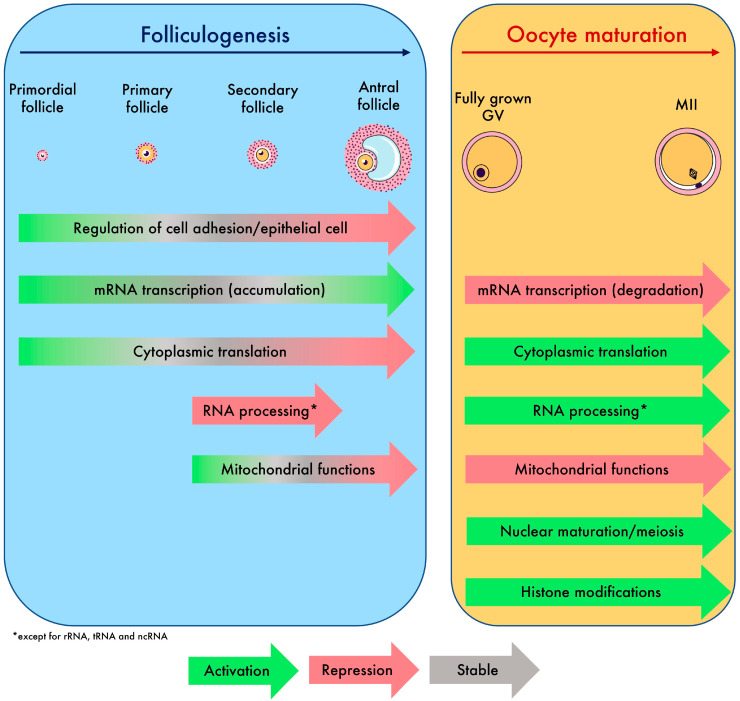
Summary of biological processes and global transcriptomic events driving oocyte growth and maturation identified in this study.

**Table 1 ijms-25-00033-t001:** Description, for each follicle stage group, of the top 5 genes differentially expressed along folliculogenesis reaching their peak expression at this stage. Their function was retrieved and simplified from Uniprot database.

Gene	Timing of Peak Expression	Full Name	Function (Uniprot Description)
*BCOR*	Primordial Primary	BCL6 Interacting Corepressor	Transcriptional corepressor May specifically inhibit gene expression when recruited to promoter regions by sequence-specific DNA-binding proteins such as BCL6 and MLLT3
*LOXL2*	Primordial Primary	Lysyl oxidase homolog 2	Mediates the post-translational oxidative deamination of lysine residues on target proteins leading to the formation of deaminated lysine Acts as a transcription corepressor and specifically mediates deamination of H3K4me3, a specific tag for epigenetic transcriptional activation
*OAZ1*	Primordial Primary	Ornithine decarboxylase antizyme 1	Ornithine decarboxylase (ODC) antizyme protein that negatively regulates ODC activity and intracellular polyamine biosynthesis and uptake in response to increased intracellular polyamine levels
*KRT19*	Primordial Primary	Keratin, type I cytoskeletal 19	Involved in the organization of myofibers
*HIST2H2AA3*	Primordial Primary	Histone H2A type 2-A	Core component of nucleosome
*AC034236.1*	Secondary		Unknown
*POMZP3*	Secondary	POM121 and ZP3 fusion protein	Unknown
*MT-CO3*	Secondary	Cytochrome c oxidase subunit 3	Component of the cytochrome c oxidase, the last enzyme in the mitochondrial electron transport chain which drives oxidative phosphorylation
*COX7B*	Secondary	Cytochrome c oxidase subunit 7B, mitochondrial	Component of the cytochrome c oxidase, the last enzyme in the mitochondrial electron transport chain which drives oxidative phosphorylation
*JUND*	Secondary	Transcription factor JunD	Transcription factor binding AP-1 sites
*TUBA1C*	Antral	Tubulin alpha-1C chain	Tubulin is the major constituent of microtubules, a cylinder consisting of laterally associated linear protofilaments composed of alpha- and beta-tubulin heterodimers
*TUBB8P10*	Antral	Tubulin Beta 8 Class VIII Pseudogene 10	Pseudogene
*ZAR1L*	Antral	Protein ZAR1-like	mRNA-binding protein required for maternal mRNA storage, translation, and degradation during oocyte maturation Probably promotes formation of some phase-separated membraneless compartment that stores maternal mRNAs in oocytes: acts by undergoing liquid–liquid phase separation upon binding to maternal mRNAs
*PCBP1*	Antral	Poly(rC)-binding protein 1	Single-stranded nucleic acid binding protein that binds preferentially to oligo dC
*WEE2*	Antral	Wee1-like protein kinase 2	Oocyte-specific protein tyrosine kinase that phosphorylates and inhibits CDK1/CDC2 and acts as a key regulator of meiosis during both prophase I and metaphase II Required to maintain meiotic arrest in oocytes during the germinal vesicle (GV) stage Also required for metaphase II exit during egg activation

**Table 2 ijms-25-00033-t002:** Comparison of folliculogenesis and oogenesis-specific genes identified in this review with previous putative genes sensitive to the environment (PCOS, maternal age, IVM, cryoconservation). Only DEGs retrieved in >2 studies for each factor were considered to increase robustness of the analysis.

Factor	Folliculogenesis Cross-Checking	GV-to-MII Cross-Checking
PCOS	*ARHGAP18 ATRX CNOT6 EEA1 KLHL32 LAMP3 LRRTM4 POMP RASA1 SCAMP2 SERPINB5 UBE2V2 ZDHHC6*	*AMOT ARHGAP18 ATRX C3orf14 CASP8 CNOT6 CTNND1 EEA1 FKBP4 NFRKB NIF3L1 PRPS2 SERPINB5 TNRC6A*
Maternal age	*ABHD5 ATRX C12orf75 C1orf146 C5orf58 CNIH4 COX7B DEAF1 GCA GPX1 HAL LGALS12 LSM8 MRFAP1 MRPL22 NDUFA1 NDUFB6 PIN1 PIR PLEKHF2 POMZP3 PTBP2 RAD23B SAA1 SCGB3A2 SLC10A3 SNAP23 STYK1 TICAM1 TMEM65 TSPAN13 TSPYL5 TUBA3E TXNDC12 UCHL3 ZAR1L*	*ABHD5 ARPC1A ARPC3 ATRX BDH2 C5orf58 CFL2 CNIH4 CYB5A DYNLL1 EIF6 ELP1 LGALS12 LSM6 MRPL22 MYL12A NDUFA1 NDUFAF6 NDUFS6 PDCD5 PIN1 PPA1 PPRC1 PSMB1 PTS RECQL4 RMI2 SCCPDH SNRPA1 SRRM1 TMEM65 UCHL3*
IVM	*ATP5MG DCTN6 LYPLAL1 NFE2L2 PAOX PCF11 TAF1A*	*DAZL DDX59 DEPDC7 EXOSC8 MKKS MRPL20 PSMB1*
Oocyte cryoconservation	*FOXO3B PRKCSH RSPH4A SF3A2 TRPC3 TYMS UBXN4 USP4 ZWINT*	*DYSF UBXN4 ZNF530*

DEGs: differentially expressed genes; IVM: in vitro maturation; PCOS: polycystic ovary syndrome.

**Table 3 ijms-25-00033-t003:** Characteristics of studies re-analyzed in this review.

Study	Year	Population	nGV	nMI	nMII	Ovarian Stimulation	Oocytes/Follicles Collection and Processing	RNA-Seq
Reyes et al. [29].	2017	*n* = 5 (1 GV and 1 IVM-MII for each patient) Age: *n* = 5 < 30 yo (26.8, 20–29/*n* = 5 >= 40 yo (41.6, 40–43) Varying causes of infertility	10		10	FSH + hCG for final follicular maturation	GV processed immediately MII obtained after 24 h IVM	Isolation: PicoPure RNA Isolation kitRT: anchored oligo(dT) Library: Thruplex DNA-seq kit Sequencing: Illumina paired-end 100 bp (HiSeq2500)
Yu et al. [26]	2020	*n* = 17 Age: GV 35.3 yo (28–41), MI 36.1 yo (32–41), MII 32.6 yo (27–39)	7	7	7	Unknown	MII from oocyte donors who had excess oocytes that removed them from donor list	RT: SMART-Seq v4 ultra low input RNA kit Library: Illumina TruSeq
Ntostis et al. [28]	2021	*n* = 12 Age: *n* = 6 21.6 yo (21–26 yo)/*n* = 6 42.0 yo (41–44 yo) Young maternal age group = oocyte donor Advanced maternal age = unexplained infertility, male infertility or age	10		11	Short GnRH agonist protocol + rFSH + hCG trigger	In vivo matured	RT: SMART-Seq v4 ultra low input RNA kit Library: Illumina’s Nextera XT Sequencing: Illumina sequencing 150 bp (HiSeq3000)
Llonch et al. [27]	2021	*n* = 37 (25 donors, 12 patients) Age: 28.8 yo (18–43) AFC: 22.1 (4–46) Patient: advanced maternal age or male factor infertility	44		31	GnRH antagonist + FSH or HP-hMG + hCG or triptorelin	GV processed immediately MII obtained after 30 h IVM	Smart-seq2 protocol RT: SuperScript II with oligo-dT Library: Illumina’s Nextera XT Sequencing: Illumina paired-end 75 bp (HiSeq4000)
Hu et al. [25]	2022	Age: <35 yo	9		5	GnRH antagonist or long agonist protocol, hCG trigger	2 replicates of 10 pooled oocytes from different donors (for GV and MII) or single oocytes Oocytes were vitrified/thawed	T&T-seq Isolation: TRIzol + isopropanol RT: Single Cell Full-Length mRNA-Amplification Kit Library: TruePrep^®^ DNA Library Prep Kit V2 Sequencing: (Novaseq 6000)
Takeuchi et al. [24]	2022	*n* = 11 Age: 30–39 yo	7	6	6		Surplus oocytes other than the MII stage were subjected to IVM: GV, MI, and MII collected after IVM Oocytes that remained immature at the time sampling were left in the maturation medium overnight and examined on the next day of the oocyte retrieval 4 GV, 3 MI, 3 MII after 7.5–9 h IVM 3 GV, 3 MI, 3 MII after 15–16.5 h IVM	RT: SMART- seq v4 Ultra Low Input RNA Kit Library: Nextera XT DNA Library Preparation Kit Sequencing: paired-end 50 bp + 25 bp (NextSeq)
			87	13	70			
Zhang et al. [23]	2018	*n* = 8 Age: 27.7 yo (24–32) Sex reassignment surgery, cervical cancer, endometrial cancer, benign ovarian mass, or lymphoma but without histopathological abnormality	*n* = 17 primordial follicles *n* = 25 primary follicles *n* = 12 secondary follicles *n* = 23 antral follicles			NA		Library: Kappa Hyper Prep Kit

AFC: antral follicle count; FSH: follicle-stimulating hormone; GnRH-a: gonadotrophin-releasing hormone; GV: germinal vesicle; hCG: human chorionic gonadotrophin; HP-hMG: highly purified human menopausal gonadotrophin; IVM: in vitro maturation; MI: metaphase I; MII: metaphase II; RT: retrotranscription.

## Data Availability

Data are contained within the article and Appendix A.

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
