# Peer review of "Overview of Gene Expression Dynamics during Human Oogenesis/Folliculogenesis"

_ijms, 2023, doi:10.3390/ijms25010033_

Round 1
Reviewer 1 Report
Comments and Suggestions for Authors
The authors' manuscript details the changes in gene expression during Human Oogenesis/Folliculogenesis, and also relates many environmental factors in a logically rigorous and narratively coherent manner.
Manuscripts with such high vertical depth are of great significance in our understanding of the whole process of oogenesis and maturation, but there are still areas that could be improved.
The authors mention in the Materials and Methods section that patients were used as the object of data analysis, but the specific results don't seem to be seen too much, and I think they could have been compared more in connection with the discussion section later in the manuscript.
On page 20, line 130, the manuscript introduces gene up- and down-regulation. The connections between amino acids as well as enzyme proteins are refreshing and appropriately laid out in the previous paragraph.
All in all, the authors' study is very meaningful and a major positive advancement for the cognitive observation of oogenesis, and the conjecture mentioned by the authors on page 21, line 173, is also very meaningful, and I very much look forward to the authors' next research.
Comments on the Quality of English LanguageThe authors' manuscript details the changes in gene expression during Human Oogenesis/Folliculogenesis, and also relates many environmental factors in a logically rigorous and narratively coherent manner.
Manuscripts with such high vertical depth are of great significance in our understanding of the whole process of oogenesis and maturation, but there are still areas that could be improved.
The authors mention in the Materials and Methods section that patients were used as the object of data analysis, but the specific results don't seem to be seen too much, and I think they could have been compared more in connection with the discussion section later in the manuscript.
On page 20, line 130, the manuscript introduces gene up- and down-regulation. The connections between amino acids as well as enzyme proteins are refreshing and appropriately laid out in the previous paragraph.
All in all, the authors' study is very meaningful and a major positive advancement for the cognitive observation of oogenesis, and the conjecture mentioned by the authors on page 21, line 173, is also very meaningful, and I very much look forward to the authors' next research.
Author Response
Reviewer 1
The authors' manuscript details the changes in gene expression during Human Oogenesis/Folliculogenesis, and also relates many environmental factors in a logically rigorous and narratively coherent manner.
Manuscripts with such high vertical depth are of great significance in our understanding of the whole process of oogenesis and maturation, but there are still areas that could be improved.
We thank Reviewer 1 for those positive comments.
The authors mention in the Materials and Methods section that patients were used as the object of data analysis, but the specific results don't seem to be seen too much, and I think they could have been compared more in connection with the discussion section later in the manuscript.
We thank Reviewer 1 for highlighting this element. It is true that differences between patients could exist due to characteristics (age, cause of infertility, …) as we discussed in the Discussion section. The Table 1 in the Results displays the characteristics of the population included in each study, which could have influenced the overall analysis. We added a sentence in the Results section to refer to this parameter for the folliculogenesis analysis : “The patients included in this study were notably healthy or suffering from reproductive pathologies of varying degrees of severity” (end of section 3.1). We also stated in line 33 that “the parameters likely to influence biological effects” of oocyte final maturation are “age or health condition” and provided information on the number of oocytes and ages of patients included in the meta-analysis : “In total, the six datasets included 87 fully grown GV and 70 MII oocytes, collected from patients ranging from young to advanced maternal ages (18-44yo).” (section 3.2).
On page 20, line 130, the manuscript introduces gene up- and down-regulation. The connections between amino acids as well as enzyme proteins are refreshing and appropriately laid out in the previous paragraph.
We thank Reviewer 1 for those positive comments.
All in all, the authors' study is very meaningful and a major positive advancement for the cognitive observation of oogenesis, and the conjecture mentioned by the authors on page 21, line 173, is also very meaningful, and I very much look forward to the authors' next research.
We are also looking forward to exploring this topic further, thanks to the discoveries detailed in this paper. We are planning several experimental studies, which we hope will enable advances in in vitro maturation techniques.

Reviewer 2 Report
Comments and Suggestions for Authors
The peer-reviewed article presents the results of a comparative analysis of previously published human single-oocyte RNA-seq datasets. The authors analyze the features of the transcriptome of developing oocytes from the primordial follicle stage to MII. Certain questions are raised by the inclusion of MII oocytes in the study, obtained as a result of IVM, as well as oocytes from donors removed from the donor list. It is very likely that such oocytes have reduced developmental abilities, which may distort the overall picture of transcriptome dynamics during oogenesis. However, the use of such material is inevitable when working with human oocytes. Considering the high demand for ART as well as their relatively low effectiveness, the topic of the article seems relevant and MS will be interesting for a wide range of specialists. Nevertheless, before publication, authors should conduct a thorough revision of the text to eliminate erroneous statements and inaccuracies.
The main points are the following:
P. 1. The authors write: “The nuclear maturation is stopped for several years and resumes after puberty at each menstrual cycle, when a surge of luteinizing hormone triggers the growth and final maturation of a dominant follicle …”.
Indeed, the peak of LG release determines the resumption and completion of the first meiotic division, but oocyte/follicle growth is regulated by follicle-stimulating hormone.
P. 2. The authors write: “Concomitantly, oocyte cytoplasmic maturation is also conditioned by two other biological modifications: organelle maturation, which situates mitochondria/ribosomes/endoplasmic reticulum/cortical granules and the Golgi apparatus, and epigenetic maturation, because important epigenetic reprogramming takes place at the same time (de novo methylation, histone modifications and exchanges) [6].”
Epigenetic reprogramming does not refer to cytoplasmic maturation, but to nuclear maturation of the oocyte.
P. 2. In the next paragraph the authors also write: “During folliculogenesis, the chromatin is decondensed, making the transcriptional activity high…. This activity is stopped when oocytes reach the GV stage…”
GV oocytes are oocytes at the diplotene stage of meiotic prophase. Thus, GV oocytes are found at all stages of folliculogenesis, including primordial and primary follicles. Complete cessation of transcriptional activity of GV oocytes occurs only at the end of the GV stage—the moment of GV breakdown (GVBD) as a result of the action of LH. What stages of folliculogenesis do the authors mean when talking about chromatin decondensation? As the tertiary follicle develops, gradual condensation of chromatin occurs, which is expressed in the formation of a characteristic heterochromatic structure around the NLB, the karyosphere.
P. 2. The combination of the terms “oocyte maturity/folliculogenesis” seems unfortunate and inappropriate. During the entire process of folliculogenesis, the oocyte grows (cf., the growth period), while maturation itself occurs during the final stages of development of the antral follicle and is completed after fertilization of the oocyte.
P. 9. From my point of view, it is incorrect to say “higher expression in MII oocytes,” since oocyte transcription stops at the onset of GVBD and resumes during minor ZGA. It is more correct to write about a higher level of transcripts.
Lines 91—93. The authors write: “First, variation in chromatin accessibility determines the activity of transcription, which is high within follicles but is stopped in fully grown GV oocytes after GV breakdown (GVDB),” whereas transcription actually stops before GVBD.
Lines 99—106. The authors write: “The primordial stage may reflect strong communication between the oocyte and its surrounding cells, manifested by an up-regulation of genes related to cell adhesion, epithelial cell proliferation, and collagen fibril organization pathways…” However, Fig. 6 shows that cell adhesion gene transcription is enhanced in antral follicles compared to primordial ones.
Fig. 4. According to Fig. 4, RNA processing is down-regulated in MII oocytes, whereas Fig. 6 demonstrates its activation in MII oocytes.
Lines 124—125. The authors write: “…mRNA processing and notably splicing are slowed down in secondary follicle oocytes compared to other stages.” How can this be explained, given that the oocyte in the secondary follicle is growing and is already actively synthesizing zona pellucida?
Lines 57—58. The authors write: “The degradation concerned transcripts from all chromosomes, whereas mtDNA was increased in MII oocytes”. Are the authors writing about mtDNA transcripts or mtDNA itself?
I didn’t find tables S1 and S2 in the zip file with supplementary materials.
Author Response
Reviewer 2
The peer-reviewed article presents the results of a comparative analysis of previously published human single-oocyte RNA-seq datasets. The authors analyze the features of the transcriptome of developing oocytes from the primordial follicle stage to MII. Certain questions are raised by the inclusion of MII oocytes in the study, obtained as a result of IVM, as well as oocytes from donors removed from the donor list. It is very likely that such oocytes have reduced developmental abilities, which may distort the overall picture of transcriptome dynamics during oogenesis. However, the use of such material is inevitable when working with human oocytes. Considering the high demand for ART as well as their relatively low effectiveness, the topic of the article seems relevant and MS will be interesting for a wide range of specialists. Nevertheless, before publication, authors should conduct a thorough revision of the text to eliminate erroneous statements and inaccuracies.
We thank Reviewer 1 for those positive comments and for highlighting elements regarding the inclusion of IVM-MII oocytes and oocytes removed from donor list which is indeed inevitable when working with human oocytes.
The main points are the following:
P.1. The authors write: “The nuclear maturation is stopped for several years and resumes after puberty at each menstrual cycle, when a surge of luteinizing hormone triggers the growth and final maturation of a dominant follicle …”.
Indeed, the peak of LG release determines the resumption and completion of the first meiotic division, but oocyte/follicle growth is regulated by follicle-stimulating hormone.
We thank Reviewer 2 for notifying us this inaccuracy. We corrected the sentence. “when a surge of luteinizing hormone triggers the final maturation of a dominant follicle”.
P.2. The authors write: “Concomitantly, oocyte cytoplasmic maturation is also conditioned by two other biological modifications: organelle maturation, which situates mitochondria/ribosomes/endoplasmic reticulum/cortical granules and the Golgi apparatus, and epigenetic maturation, because important epigenetic reprogramming takes place at the same time (de novo methylation, histone modifications and exchanges) [6].”
Epigenetic reprogramming does not refer to cytoplasmic maturation, but to nuclear maturation of the oocyte.
We agree with Reviewer 2 and we have replaced “cytoplasmic” to “nuclear”.
P.2. In the next paragraph the authors also write: “During folliculogenesis, the chromatin is decondensed, making the transcriptional activity high…. This activity is stopped when oocytes reach the GV stage…”
GV oocytes are oocytes at the diplotene stage of meiotic prophase. Thus, GV oocytes are found at all stages of folliculogenesis, including primordial and primary follicles. Complete cessation of transcriptional activity of GV oocytes occurs only at the end of the GV stage—the moment of GV breakdown (GVBD) as a result of the action of LH. What stages of folliculogenesis do the authors mean when talking about chromatin decondensation? As the tertiary follicle develops, gradual condensation of chromatin occurs, which is expressed in the formation of a characteristic heterochromatic structure around the NLB, the karyosphere.
We thank Reviewer 2 for notifying us of this omission, we were indeed referring to fully grown GV oocytes at the moment of GVBD. We have also corrected the reference to the chromatin condensation during folliculogenesis, which is decondensed in pre-antral follicles and starts to gradually condense in tertiary follicles, as reported by the Reviewer 2 : “In the pre-antral follicles, the chromatin is decondensed, making the transcriptional activity high and fitting the needs required by the follicle to grow [7,8]. As the tertiary follicle develops, gradual condensation of chromatin occurs, and the transcriptional activity is finally stopped when oocytes reach the GV breakdown (GVBD) and could initiate drastic mRNA degradation”.
P.2. The combination of the terms “oocyte maturity/folliculogenesis” seems unfortunate and inappropriate. During the entire process of folliculogenesis, the oocyte grows (cf., the growth period), while maturation itself occurs during the final stages of development of the antral follicle and is completed after fertilization of the oocyte.
We have modified this formatting to separate oocyte maturation and folliculogenesis.
P.9. From my point of view, it is incorrect to say “higher expression in MII oocytes,” since oocyte transcription stops at the onset of GVBD and resumes during minor ZGA. It is more correct to write about a higher level of transcripts.
We have proceeded to the modifications suggested by Reviewer 2.
Lines 91—93. The authors write: “First, variation in chromatin accessibility determines the activity of transcription, which is high within follicles but is stopped in fully grown GV oocytes after GV breakdown (GVDB),” whereas transcription actually stops before GVBD.
We have modified our choice of terms in this sentence to refer to “early follicles” and that the transcriptional activity is “absent” after GVBD (line 95).
Lines 99—106. The authors write: “The primordial stage may reflect strong communication between the oocyte and its surrounding cells, manifested by an up-regulation of genes related to cell adhesion, epithelial cell proliferation, and collagen fibril organization pathways…” However, Fig. 6 shows that cell adhesion gene transcription is enhanced in antral follicles compared to primordial ones.
We thank Reviewer 2 for highlighting an issue in the colors used in the Figure 6. We have corrected the colors in the Figure 6, accordingly.
Fig. 4. According to Fig. 4, RNA processing is down-regulated in MII oocytes, whereas Fig. 6 demonstrates its activation in MII oocytes.
We thank Reviewer 2 for pointing out an ambiguity regarding Figures 4 and 6. Indeed, Figure 4 indicates the majority of RNA processes are activated in MII compared to fully grown GV oocytes (framed in red) but some RNA processes related to rRNA, tRNA and ncRNA are down-regulated in MII oocytes. We specified in Figure 6 that RNA processes are activated in MII oocytes except for rRNA, tRNA and ncRNA.
Lines 124—125. The authors write: “…mRNA processing and notably splicing are slowed down in secondary follicle oocytes compared to other stages.” How can this be explained, given that the oocyte in the secondary follicle is growing and is already actively synthesizing zona pellucida?
This is an interesting question. This observation may arise because of the kind of analysis we performed, which compares one follicle stage group to the three other follicle groups. Thus, these processes may still be maintained at a high level of activity to fit the needs of the oocyte but appear to be down-regulated in secondary follicles, because compared to the other three groups and, notably the antral group, the transcripts participating in these processes appeared a little down-regulated. So, there may be a general slow-down in such processes compared to other follicles, which does not compromise oocyte growth. Besides, growth may be driven by the expression of growth factors such as FIGLA, which is high in oocytes in secondary follicles oocytes (logFC = 1.08, adj.p = 0.02) compared to other follicle stage oocytes. We can also see that ZP1 and ZP3 are highly expressed during the secondary follicle stage (logFC = 2.88 and 1.06, adj.p=0.0003 and 0.02 respectively), indicating that the zona pellucida synthesis is not compromised and the correct transcriptional program is followed inside the oocyte.
Lines 57—58. The authors write: “The degradation concerned transcripts from all chromosomes, whereas mtDNA was increased in MII oocytes”. Are the authors writing about mtDNA transcripts or mtDNA itself?
We refer to transcripts of the mtDNA-encoded genes. We have corrected this designation (“mtDNA gene expression”).
I didn’t find tables S1 and S2 in the zip file with supplementary materials.
We apologize for the omission of Supplementary Tables in the submitted version. They are now available in the Supplementary Material zip file.

Reviewer 3 Report
Comments and Suggestions for Authors
The topic of the present paper Overview of Gene Expression Dynamics during Human Oogenesis/Folliculogenesis is very interesting for readers, because the authors first applied a Gene Ontology and a pseudotime analysis to identify the fundamental transcriptomic requirements of the oocyte at any stage of folliculogenesis and the resultant mRNA regulation.
The authors performed a meta-analysis of studies comparing fully-grown germinal vesicle stage and the metaphase II stage oocytes to decode the accumulation/degradation of maternal transcripts occurring during this final maturation. Furthermore, the present study evaluated whether the transcripts supposedly having a role in the oocyte growth and maturation processes are sensitive to environmental factors, which could help to a better understand the origin of oocyte maturation defects.
The authors concluded that the oocyte transcriptome follows a dynamic that is tightly controlled. The atlas of transcriptomic modifications described in the present manuscript will facilitate the precise identification of the transcripts involved in the failure of oocyte growth and maturation in future research.
So, finally I conclude that:
- the introduction provides sufficient background and includes relevant references;
- the reference list is large and recently;
- the manuscript is well written, and the text is easy to read.
Author Response
Reviewer 3
The topic of the present paper Overview of Gene Expression Dynamics during Human Oogenesis/Folliculogenesis is very interesting for readers, because the authors first applied a Gene Ontology and a pseudotime analysis to identify the fundamental transcriptomic requirements of the oocyte at any stage of folliculogenesis and the resultant mRNA regulation.
The authors performed a meta-analysis of studies comparing fully-grown germinal vesicle stage and the metaphase II stage oocytes to decode the accumulation/degradation of maternal transcripts occurring during this final maturation. Furthermore, the present study evaluated whether the transcripts supposedly having a role in the oocyte growth and maturation processes are sensitive to environmental factors, which could help to a better understand the origin of oocyte maturation defects.
The authors concluded that the oocyte transcriptome follows a dynamic that is tightly controlled. The atlas of transcriptomic modifications described in the present manuscript will facilitate the precise identification of the transcripts involved in the failure of oocyte growth and maturation in future research.
So, finally I conclude that:
- the introduction provides sufficient background and includes relevant references;
- the reference list is large and recently;
- the manuscript is well written, and the text is easy to read.
We thank Reviewer 3 for those positive comments.
